# Spontaneous crack healing in calcite reveals the influence of dynamic strain evolution and surface chemistry

Michelle Devoe [1,4] ✉, Harrison P. Lisabeth [2,4], Seiji Nakagawa[2], Zhao Hao [2], Nobumichi Tamura [3] & Hans-Rudolf Wenk[1]

The mechanics of fracture healing in calcite remain poorly constrained yet are fundamental to managing fluid transport in geothermal reservoirs and hydrocarbon systems. Here, we apply microfocused synchrotron Laue X-ray diffraction and infrared spectroscopy to investigate subcritical crack healing in a 1 mm-thick calcite crystal subjected to controlled loading in a double-torsion device. Over a 44-hour period following load removal, we map the evolution of residual strain fields surrounding the crack tip and observe a progressive increase in compressive strain perpendicular to the crack plane accompanied by infrared spectroscopic signatures that reveal enhanced accumulation of water at the healed interface. The correlation between strain evolution and surface chemistry suggests that spontaneous crack healing in calcite is driven by dynamic anelastic relaxation coupled with irreversible fluid-mineral interactions. These findings offer insight into time-dependent crack closure processes in carbonates and highlight the role of chemically-mediated plasticity in subsurface fracture evolution.

Subcritical crack growth (SCG) and spontaneous crack healing (SCH) are governed by a complex interplay of stress corrosion, dissolution-precipitation, ion exchange, microplasticity, and residual strain, requiring multi-scale and multi-modal approaches to decipher their evolution. While the crack mechanics of ductile and metallic systems are well-characterized, where microstructural damage ahead of the crack tip competes with shielding processes behind[1,2], the corresponding behavior in macroscopically brittle geologic materials, those which deform brittlely but may accommodate some crystal plasticity at the grain scale, remains poorly resolved and the influence of crystal plasticity, residual stress, and environment-driven anelasticity on crack closure is only beginning to be explored.

The deformation and fracture behavior of calcite (CaCO₃) and calcite-rich rocks such as marble have been investigated experimentally for over a century[3–6], owing to the mineral's abundance in the Earth's crust and its ability to deform readily under low confining pressures.

Calcite remains relevant due to its prevalence in hydrocarbon basins and geothermal reservoirs, in crustal fault zones, as well as its ubiquity and influence on the mechanical properties of engineered materials like concrete[7–12]. Fluid transport in energy-rich hydrocarbon basins and geothermal reservoirs is governed by the competing processes of crack propagation and crack healing in the reservoir's rock formations[13]. The connectivity of fracture networks to the wellbore, either inherent to the rock formation or artificially created through reservoir stimulation, determines the delivery rate and overall economic recovery of the reservoir[14–16]. Permeability of the formation changes over time due to continuous fluid-rock interactions and mineral precipitation, such as crack sealing processes where secondary minerals from distant sources precipitate onto the fracture surfaces, potentially occluding the fracture and requiring stimulation to improve reservoir permeability[17–19]. Crack healing, the process in which the fracture surface is eliminated, and

[1]Department of Earth and Planetary Science, University of California, Berkeley, CA, USA. [2]Energy Geoscience Division, EESA, Lawrence Berkeley National Laboratory, Berkeley, CA, USA. [3]Advanced Light Source, Lawrence Berkeley National Laboratory, Berkeley, CA, USA. [4]These authors contributed equally: Michelle Devoe, Harrison P. Lisabeth. ✉e-mail: mdevoe@berkeley.edu

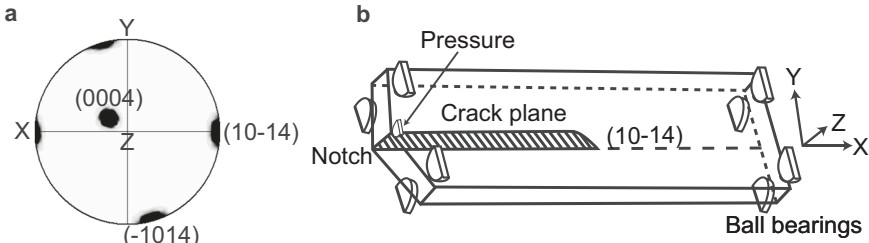

**Fig. 1 | Sample orientation and DT schematic. a** Pole figure showing the orientation of calcite cleavage planes relative to the sample coordinate system (X, Y, Z). The cleavage plane is aligned parallel to the *X* axis. Equal-area projection. **b** Schematic of the custom double-torsion device used to propagate a controlled Mode I crack along the (10–14 plane of a pre-notched calcite slice. Load is applied via a pin fitted with a ball-bearing tip, which exerts pressure directly above the notch, initiating tensile cracking that splits the specimen into two domains. These domains behave as independent torsion bars under load, giving rise to the term "double-torsion." In this modified configuration, four additional ball bearings are used on the top side of the slice to securely hold the sample in position during X-ray diffraction measurements. Sample coordinate axes are indicated.

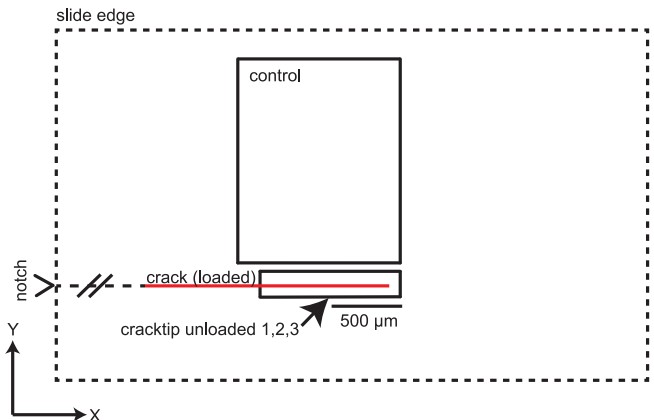

**Fig. 2 | Map schematic showing relative scan positions.** Spatial relationships of Laue diffraction scans collected during the experiment. The solid red line represents the location of the crack under load relative to the cracktip_unloaded scans. Note that the distance between the calcite slide edge (dashed black line) and the scan areas (solid black lines) is not to scale. Sample coordinate system (X, Y).

the lattice is repaired via mass transport operations like diffusion, is usually aided by factors such as heat, time, and confining pressure[17,18,20]. Distortion, displacement, and contamination of the fracture surface can prevent strong bonds from forming between the two surfaces and reduce the strength of re-healing[21]. A perfectly healed crack would demonstrate strength equivalent to its pre-crack state[21]. SCG, the time-dependent propagation of cracks at stress intensities below the critical threshold[22], has been identified as a key driver of carbonate sediment compaction, fracture size and spacing, and fracture propagation velocity, which can affect reservoir permeability, and crustal fault zone stability (e.g., [9,22–26]). Conversely, SCH, whereby cracks gradually close and mechanical properties partially recover once the driving force of propagation is removed, remains less well understood despite its potential to significantly alter permeability and seismic stability[20,27].

Understanding the mechanisms driving SCG and SCH is essential for maintaining the production performance of energy-rich reservoirs and predicting the mechanical stability of fault-zone carbonate-rich rock formations.

To address this, we apply a time-resolved, synchrotron-based approach to directly observe strain evolution during crack healing in a single crystal of calcite. Using a double-torsion apparatus, we propagated a controlled Mode I crack in a calcite slide, then monitored the real-time redistribution of deviatoric strain over the course of the 44 hours following load removal using microfocused synchrotron Laue X-ray diffraction at the Advanced Light Source (ALS) beamline 12.3.2. This technique enables high-resolution mapping of elastic strain within bulk samples, as previously demonstrated with metallic thin films[28], rolled titanium[29], and polycrystalline geologic materials[30], and offers a unique window into the post-fracture relaxation processes operating at the micro-scale. Infrared spectroscopy of the crack plane after load removal was used to assess chemical alteration of the crack plane, capturing irreversible surface reactions.

This work sheds light on the physical and chemical drivers of fracture closure in a common reservoir mineral. The results reveal how mechanical and environmental interactions co-evolve in fractured carbonates and underscore the importance of dynamic residual stress and surface chemistry in governing porosity and permeability evolution across a range of geologic settings.

## Results

### Stress evolution measured with Laue microdiffraction

A pre-notched, rectangular slice of calcite measuring 30 mm × 10 mm × 1 mm was used in this experiment. A custom double-torsion device was used to propagate a Mode I crack parallel to the (10-14) cleavage plane using a small ball-bearing placed at the tip of a cup-tipped set screw, which applied a controlled 3-point bending force directly over the notch (Fig. 1a, b). This design held the calcite slice perpendicular to the incident beam (Supplementary Fig. 1a) and maintained constant load, which enabled precise control over crack growth under Mode I loading conditions. The sample was then partially unloaded, and elastic strain maps around the crack were collected by translating the sample stage around the X and Y coordinates in a stepwise manner and capturing a Laue diffraction image at each step. The step sizes, or the length of translational stage displacement, varied depending on the purpose of each scan. A shadow from the device can be observed on the right-most panels of the detector (Supplementary Fig. 1b). Details on the experimental set-up and beamline geometry are provided in Section 5.

Prior to crack propagation, a control scan (1.5 mm × 2 mm) was collected to establish a baseline strain field (control, Fig. 2). The purpose of the control scan was to collect baseline strain values of the sample prior to crack propagation and was intended to overlap with the path of crack propagation. However, the control scan was misplaced and was collected slightly above the ultimate location of the crack plane due to the inability to predict its propagation path. The sample was then manually loaded using the ball-bearing-tipped actuator, propagating a fresh tensile crack ~15 mm in length along the (10-14)plane. From here on, the "crack tip" refers to the optically visible leading edge of the crack plane. For follow-up scans, permanent ink fiducial markers were applied to the sample surface to mark the

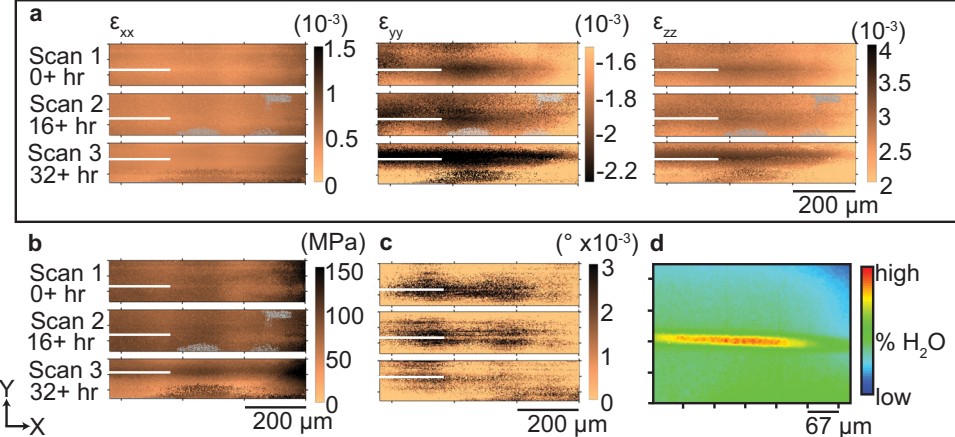

**Fig. 3 | Strain, diffraction peak width, and IR maps. a** Maps of deviatoric strain components ($\varepsilon_{xx}$, $\varepsilon_{yy}$, $\varepsilon_{zz}$) from the crack healing portion of the experiment: scans 1 (top, 0 hrs), 2 (middle, 17+ hrs), and 3 (bottom, 34+ hrs). For each strain component, the minimum value from the pre-crack control scan was subtracted to normalize the initial stress state, and the corrected strain values are shown. The white line is added only as a visual aid to mark the vertical coordinate location of the crack plane (visible while under load), and its length is arbitrary. Gray pixels represent regions where diffraction peaks could not be indexed and were excluded from the analysis. **b** Von Mises stress maps (in MPa) for the healing portion of the experiment: scans 1,

2, and 3, showing the evolution of equivalent stress in the former region of the crack tip (while under load) following unloading. **c** Maps of the average diffraction peak width residual for the healing portion of the experiment: scans 1, 2, and 3, illustrating spatial variations in crystallographic distortion associated with plastic deformation. In sample (X, Y, Z) coordinates, X-direction is parallel to [10−14] cleavage direction, and Y- and Z- directions are orthogonal. **d** Infrared spectroscopy map showing relative water concentration around healed crack after load removal. Red is high, blue is low water percentage. Horizontal and vertical scales are equivalent.

location of the crack tip under load; its approximate location relative to the cracktip_unloaded scan dimensions is visualized in Fig. 2. The load was then reduced. Upon partial unloading, spontaneous crack closure was observed: the crack tip retreated along the plane to the left towards the pre-notched edge of the sample, and the former location of the crack tip became optically indistinguishable from the surrounding crystal, indicating a self-healing response. To investigate the evolution of strain during this healing process, three high-resolution Laue microdiffraction scans (cracktip_unloaded 1, 2, 3; Fig. 2) of the former location of the crack tip were collected. During data collection, the crack tip was located to the left of the cracktip_unloaded scans and was not included in the scan area. Each cracktip_unloaded scan measured ~0.17 mm × 1.4 mm, with a reduced vertical range in cracktip_unloaded 3 (shortened by 50 μm) to accommodate time constraints. The approximate location of the crack tip (while under load) relative to the cracktip_unloaded scans 1, 2, and 3 is visualized in Fig. 2.

The step size, or stage displacement lengths between diffraction images collected, varied depending on the purpose of the scan. The control scans used large step sizes (10 μm) to cover a larger region at lower resolution. In contrast, the cracktip_unloaded scans used fine step spacing (2 μm) to generate high-resolution strain maps around the former location of the crack tip. Due to the small step size, these datasets were time-intensive: cracktip_unloaded 1 and 2 each collected ~59,000 diffraction images over ~16 hours, and unloaded 3 collected ~45,000 patterns in 12 hours. As such, each scan should be considered a temporally distributed dataset, with several hours elapsing between the first and last diffraction image collected within each map. Scan measurements were not corrected for temporal artifacts as time-dependent slip system activity could not be resolved, and strain rates could not be quantified.

Deviatoric strain tensors were used to quantify the elastic strain fields from each Laue diffraction image using the XMAS software package[31]. For more details on how strain is calculated, the reader is referred to[29,31,32]. Following crystallographic indexing of diffraction peaks, the local deviatoric strain tensor was calculated by comparing observed peak positions with those expected for an ideal, unstrained calcite lattice. Because deviatoric strain quantifies relative changes in

the shape of the unit cell, the sign convention assigned is arbitrary. In this study, positive values are associated with strain values indicating an extension, and negative values are associated with strain values indicating a contraction as compared to an ideal, unstrained calcite lattice. An Eigen decomposition converts the deviatoric strain tensor from a coordinate system attached to the unit cell into the coordinate system of the sample ($\varepsilon$).

The corresponding stress tensor ($\sigma$) was then derived by applying Hooke's law ($\sigma = \varepsilon \cdot C$), using the stiffness tensor for calcite ($C$)[33]. In addition, von Mises stress was computed to represent a scalar equivalent stress magnitude incorporating both normal and shear components (Eq. 1)[34].

$$\sigma_{vm} = \sqrt{\frac{(\sigma_{11} - \sigma_{22})^2 + (\sigma_{22} - \sigma_{33})^2 + (\sigma_{33} - \sigma_{11})^2 + 6(\sigma_{12}^2 + \sigma_{23}^2 + \sigma_{31}^2)}{2}} \quad (1)$$

To isolate the effects of crack propagation and healing and remove any inherent strain in the slide, all strain and stress data were baseline-corrected by subtracting the minimum value measured in the pre-crack control scan for each field (Supplementary Table 1).

Upon unloading, spatially diffuse negative strain in the Y-direction ($\varepsilon_{yy}$), perpendicular to the crack plane, was initially observed in scan 1, but intensified and became more localized along the crack plane in scans 2 and 3 (Fig. 3a). This progressive increase in negative strain in the Y-direction ($\varepsilon_{yy}$) was accompanied by the accumulation of positive strain in the Z-direction ($\varepsilon_{zz}$), parallel to the X-ray beam path and along the thickness of the sample. The coherent, uniform pattern of strain in the Z-direction indicates limited dilatational strain, which means the negative strain in the Y-direction can be interpreted as compression. Strain in the X-direction ($\varepsilon_{xx}$), parallel with the path of crack propagation, remained minimal and largely invariant throughout the post-unloading scans. These trends suggest the development of a dynamic strain field in the region of the former crack tip after the load had been removed that effectively closes the fracture.

Quantitatively, the minimum compressive strain perpendicular to the crack ($\varepsilon_{yy}$) became increasingly more negative over time, decreasing from −2.428 millistrain in scan 1 to −3.041 millistrain in scan

3. Concurrently, the maximum strain value perpendicular to the crack ($\varepsilon_{yy}$) became less negative, increasing from −1.033 to 0.2790 millistrain between scans 2 and 3 (Table 1). This reflects the time-dependent accumulation of strain along the crack plane and simultaneous relaxation in the surrounding material. A similar trend was observed for $\varepsilon_{zz}$: the maximum tensile strain values increased from 3.417 millistrain in scan 1–3.796 millistrain in scan 3, while the minimum strain values decreased from 1.464 to 0.076 millistrain, suggesting both accumulation and redistribution of tensile strain along the crack plane.

Von Mises stress, which provides a scalar representation of the overall stress state, also increased along the crack plane during the post-unloading period. Maximum values rose from 174 MPa in scan 1 to 258 MPa in scan 3, while minimum values declined from 50 MPa to near zero, indicating stress accumulation around the former region of the crack tip (prior to unloading) and concurrent relaxation in the surrounding material (Fig. 3b). This temporal evolution mirrors that observed in the deviatoric strain components.

### Localized plasticity measured using diffraction peak width

Dislocation density and lattice defect accumulation were evaluated using Laue diffraction peak broadening[35]. To isolate peak broadening caused by plastic deformation from that caused by variation in X-ray beam intensity, the peak broadening due to X-ray beam intensity was subtracted from the average peak width value of each diffraction image. Trendlines correlating the measured average peak width value from each diffraction image to X-ray beam intensity, which varied over the course of each scan due to decreasing accelerator conditions during scans 2 and 3, were used to calculate the contribution of peak width caused by the variable X-ray beam intensity (Supplementary Fig. 2). The residual value between the calculated and measured average peak width value for each diffraction image was plotted (Fig. 3c) and used to observe the evolution of plastic strain around the crack. An increase in peak width is observed in a two-lobed pattern around the crack plane in scan 1. A planar feature of lesser average peak width coincident with the crack plane can be observed. This feature widens in the Y-direction and becomes more obvious in scan 2. By scan 3, the area of increased average peak width has shrunk to the immediate crack surroundings.

Dislocation activity along four common slip systems in calcite− {10-14}⟨1-210⟩, r-slip {10-14}⟨-2021⟩, and f-slip {01-12}⟨0-111⟩ and {01-12}⟨-2021⟩−was evaluated using the dislocation study tool of PYXIS software[36–39]. Elongation of diffraction peak profiles reflects dislocation activity because lattice distortions caused by dislocations introduce anisotropic strain fields, which broaden and elongate the intensity distribution of the peak in the direction of the active slip system[39]. The diffraction image from the same stage coordinate position near the crack plane of unloaded scans 1, 2, and 3 was used to compare slip system activity during crystal healing (Fig. 4a). Because the control scan area ultimately did not overlap with the crack tip scan area, a diffraction image from the control scan was used with the same X-stage coordinate but with a Y-coordinate 3 mm above the crack. The orientation of the long axis of the diffraction peak is compared against the expected orientation for the selected slip system (Fig. 4b−d). Only slip systems with a long axis orientation variance (Δ°) of less than 10° were included in our analysis. Results show that the common four slip systems were active during unloading, and slip system activity across scans 1, 2, and 3 appears to be consistent with time elapsed since unloading (Table 2). Peak shape evolution is most prominent between the control scan and the first cracktip unloaded scan (Fig. 5). Systematic variations in peak shape occur during cracktip unloaded scans 1, 2 and 3; most notably, peak elongation in the vertical direction decreases with increased time post load removal; however, variation is minor within the resolution of the diffraction image. Only one diffraction image was analyzed using PYXIS due to the time required for the analysis. For this reason, the spatial distribution of slip system

**Table 1 | Minimum and maximum strain values**

| | $\varepsilon_{xx}(10^{-3})$ | | | $\varepsilon_{yy}(10^{-3})$ | | | $\varepsilon_{zz}(10^{-3})$ | | | Von Mises stress (MPa) | | |
|---|---|---|---|---|---|---|---|---|---|---|---|---|
| | Min. | Max. | Avg. ± Std. Dev. | Min. | Max. | Avg. ± Std. Dev. | Min. | Max. | Avg. ±Std. Dev. | Min. | Max. | Std. Dev. |
| Scan 1 | 0.2645 | 0.9870 | 0.5077 ± 0.1140 | −2.428 | −0.9572 | −1.780 ± 0.1907 | 1.464 | 3.417 | 2.614 ± 0.2800 | 42 | 174 | 85 ± 17 |
| Scan 2 | 0.3121 | 1.041 | 0.5473 ± 0.1496 | −2.614 | −1.033 | −1.866 ± 0.4373 | 1.357 | 3.545 | 2.660 ± 0.6167 | 50 | 172 | 89 ± 23 |
| Scan 3 | 0.2868 | 1.432 | 0.5668 ± 0.1597 | −3.041 | 0.2790 | −1.792 ± 0.5422 | 0.0763 | 3.796 | 2.567 ± 0.6299 | 0 | 258 | 79 ± 29 |

Minimum and maximum values of the baseline-corrected deviatoric strain components ($\varepsilon_{xx}$, $\varepsilon_{yy}$, and $\varepsilon_{zz}$) and von Mises stress measured in the cracktip_unloaded scans 1, 2, and 3. These values reflect the progressive evolution of strain and stress fields in the region of the former crack tip following load removal.

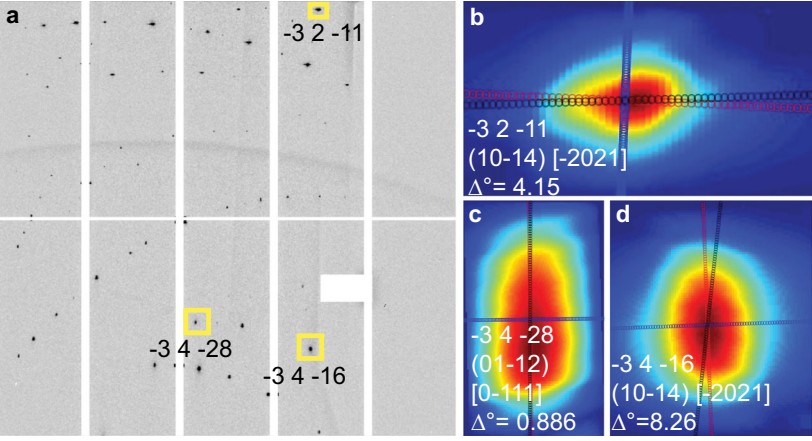

**Fig. 4 | PYXIS slip system analysis. a** A typical diffraction image from the crack plane area following unloading (cracktip_unloaded3). Three diffraction peaks, boxed in yellow, have been labeled with their Miller indices (*hkl*). PYXIS dislocation analysis of peak **b** -3 4 -28 and slip system (01-12) [0-111], **c** -3 4 -16 and slip system (10-14) [-2021], and **d** -3 2 -11 and slip system (10-4) [-2021] showing short axis (blue), long axis (red), and slip system elongation direction (black, overlain). Angular variation between the slip system elongation direction and the long axis of the peak is shown (Δ°). Intensity ranges from minimum (blue) to maximum (red).

## Table 2 | Slip system activity

| Peak (*hkl*) | Slip system (*hkil*) | | | |
|---|---|---|---|---|
| | (10–14) [–2021] | (10–14) [1–210] | (01–12) [–2021] | (01–12) [0–111] |
| -1 1 -14 | 1 | 1 | 1 | 1 |
| -2 1 -12 | - | 1 | - | - |
| -3 2 -11 | 1, 2, 3 | - | 1, 2, 3 | 1, 2, 3 |
| -3 4 -16 | 1, 3 | - | 1, 2, 3 | 1, 3 |
| -3 4 -28 | 1, 2, 3 | - | - | 1, 2, 3 |
| -4 1 -32 | - | - | - | - |

Slip system activity identified for selected diffraction peaks (*hkl*) in scans 1, 2, and 3. For each peak, the scan number in which a given slip system was detected is indicated. Note: Miller indices (*hkl*) were assigned by the PYXIS software during peak indexing, while slip systems are reported in the four-index Miller-Bravais (*hkil*) notation, which is standard for describing crystallographic planes and directions in trigonal systems such as calcite.

activity across the scan was not evaluated. Additionally, a dominant deformation process for the experiment could not be determined from this analysis due to the limited number of slip systems, peaks, and diffraction images used in the analysis. Generally, slip systems with lower critical resolved shear stress are more easily activated. Slip systems that were identified as active for more than one scan suggest they could be a more dominant deformation mechanism than others, however in the diffraction image analyzed only four possible slip systems were studied, and neither the r- or f- systems were observed as significantly more active than the other (Table 2), thus a dominant deformation mechanism for the system cannot be identified. Further study into a larger population of peaks and slip systems across multiple diffraction images could yield greater insight into the deformation process during spontaneous healing.

### Water identified around crack plane using IR spectroscopy

Infrared spectroscopy, performed after X-ray diffraction data collection and removal of the calcite slide from the double-torsion device, revealed a persistent, localized accumulation of water along the healed crack interface. As water was not added to the experiment, its presence is attributed to ambient humidity. To assess the spatial distribution of water, we conducted FTIR hyperspectral imaging and generated semi-quantitative maps of water content by integrating the absorption intensity of the O−H bending mode centered at ~1620 cm$^{-1}$ or at a

wavelength of ~6.2 μm (Fig. 3d). This vibrational mode is a reliable proxy for molecular water[40] and enables spatially resolved, diffraction-limited imaging of hydration state using the Beer-Lambert law, which relates absorption intensity to concentration through known optical parameters.

Notably, the water signal was not uniformly distributed across the sample surface but was instead tightly localized to a narrow, 15 μm-wide band centered along the location of the crack tip and plane, which were visible prior to unloading. The signal extended ~7 μm into the bulk from the crack interface. This spatial pattern indicates that water accumulation is not limited to the crack surface, but reflects chemical modification of the subsurface region below the crack surface, albeit limited by the IR attenuation depth on the order of several micrometers. With 14–16 hours elapsed between the first and last diffraction image collected for each scan, interpretation of the asymmetric shape of the water halo above and below the crack plane, which mimics the shape of the deviatoric strain (Fig. 3a), is limited to possibly an artefact due to this temporal distortion. The persistence of the water signal—despite prior scanning electron microscopy (SEM) under high vacuum and prolonged ambient storage—suggests that the water is strongly retained, either via chemisorption or incorporation into defect structures. Prior studies have shown that molecular water can remain bound to calcite surfaces, particularly the (10-14) cleavage plane, even under high-vacuum conditions[41,42], and that one such water layer was measured to be 5–6 nm thick[43], an order of magnitude thinner than the water signal measured in this study, supporting the hypothesis that water has migrated into the bulk of the material.

If the observed water distribution does indeed result from diffusion, the calculated apparent diffusivity would exceed expected values for room-temperature diffusion in pristine calcite[44], implying either enhanced transport along dislocation networks[45] or fast pathways created by microstructural damage[46]. Additional experimental work is needed to identify the mechanism of water emplacement, whether adsorption, absorption, or diffusion. Because the fracture zone follows crack front geometry and may transect the thickness of the sample (Supp. Fig. 3), water adsorption along the microcrack surfaces would be expected to be present throughout the entire thickness of the sample so long as the microcrack network is connected to the crack-air interface. Due to the shallow depth sensitivity of the IR method used in this study, surface adsorption and condensation along the crack surface, followed by mass transport along microcracks, or areas of high free energy such as defect sites, are the most plausible mechanisms[47].

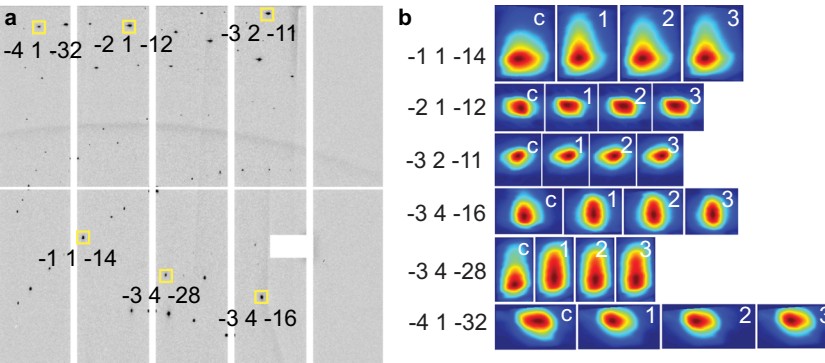

**Fig. 5 | Peak shape evolution following unloading. a** Typical diffraction image from the crack plane area following unloading (cracktip_unloaded3). Select diffraction peaks (*hkl*), boxed in yellow, are labeled and correspond to those shown in (**b**). **b** Peak shape evolution from the control scan (before loading) (labeled "c"), through cracktip_unloaded scans 1 ("1"), 2 ("2"), and 3 ("3"), from left to right. Intensity ranges from minimum (blue) to maximum (red).

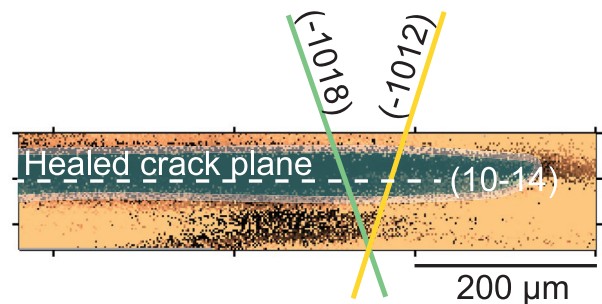

**Fig. 6 | Strain evolution schematic.** Schematic showing the relative orientations of cleavage plane (10-14), glide plane (-1012), and twin plane (-1018) overlain on the $\varepsilon_{yy}$ deviatoric strain data from scan 3. The white dashed line shows the interpretation of the "healed", or optically indistinguishable, crack plane following unloading. Blue overlay shows an interpretation of the approximate area of water observed in the IR data overlain on the deviatoric strain.

In either case, the persistent presence of water reflects a permanent chemical alteration of the fracture zone potentially linked to the observed residual strain and plastic deformation (Fig. 6).

Moreover, the retention of water in the healed region despite the apparent optical closure of the crack suggests an irreversible change in the local hydration state of the crystal. This bound water may contribute to the strain anomalies observed near the crack, as water incorporation into defect-rich regions can locally swell the lattice or modify surface energy. Importantly, such chemically modified zones may remain mechanically weaker than the pristine crystal, thereby impeding full recovery of tensile strength and stiffness[48,49]. The combination of residual water, persistent plasticity, and strain gradients indicates that spontaneous crack closure does not equate to mechanical restoration, and that water plays a critical role in modulating both surface reactivity and fracture mechanics in calcite.

## Discussion
Following the propagation of a tensile crack in single-crystal calcite and subsequent unloading, our results revealed an evolving residual stress field centered around the region of the former crack tip that became optically indistinguishable from the bulk following load removal. Over time, compressive strain in the direction perpendicular to the crack plane ($\varepsilon_{yy}$) and tensile strain co-planar with the crack plane ($\varepsilon_{zz}$) increased in magnitude and localized closer to the location of the crack-air interface. These changes corresponded to an overall accumulation of von Mises stress near the crack and a decrease elsewhere in the scanned region. The directionality and spatial coherence of these evolving stress fields are consistent with the accumulation of

plastic deformation, potentially mediated by dislocation activity and surface interaction with water.

In double-torsion tests, crack fronts have a curved geometry, with the crack front extending further on the tensile side than on the compressive side (Suppl. Fig. 3). This asymmetry can produce variability in parameters along the profile of the crack front[50]. In the reconciliation of the datasets, it is important to consider that the strain measured from Laue diffraction images represents bulk-averaged strain due to complete penetration through the sample thickness by the X-ray beam whereas IR measurements probe only the top several micrometres of the sample surface. Due to the curvature of the crack front, X-ray paths closer to the crack tip (Path B, Suppl. Fig. 3) will probe overall a lower volume of distorted lattice than areas where the crack fully propagated (Path A, Suppl. Fig. 3). This effect may be the reason the $\varepsilon_{yy}$ and $\varepsilon_{zz}$ strain magnitudes on right side of scan 1 (Fig. 2a), where the crack tip was formerly located prior to unloading, are less than strain magnitudes to the left of scan 1, through lattice which the crack fully propagated. This difference in strain magnitude between the right and left sides of the scan diminishes in scans 2 and 3, potentially due to the reconfiguration of strain as time elapses.

Increases in diffraction peak width, especially concentrated along the crack plane, are indicative of defect accumulation—namely, dislocations and localized plasticity[35]. The presence of broadened peaks decreased significantly from scan 2 to scan 3, which, when considered alongside the growing compressive strain perpendicular to the crack, supports the interpretation that dislocations are migrating toward the fracture interface and being annihilated or absorbed at the surface. This behavior aligns with dislocation recovery in annealed calcite, where climb and annihilation mechanisms progressively reduce lattice strain during relaxation[51]. These observations suggest that dislocation mobility can occur in calcite under low-stress regimes, provided the residual stress fields near the crack surface provide sufficient driving force to overcome Peierls barriers[52,53].

Diffraction peak shape analysis identified r-slip (10-14) and f-slip (01-12) system activation, with orientations favorable for migration toward the crack surface. These results are consistent with prior studies of torsional and compressive deformation in calcite, where dislocation glide was observed along similar planes[54,55]. The presence of dislocation pathways intersecting the fracture provides a plausible mechanism for lattice defects to be funneled toward and released at the crack tip, facilitating stress relief and physical crack closure (Fig. 6). Similar mechanisms have been observed in other geological materials such as olivine, where low-temperature plasticity and strain hardening emerge from persistent dislocation interactions[56]. In both cases, localized strain triggers time-dependent recovery and strengthening

behaviors previously thought to be limited to high-temperature ductile regimes.

Scanning infrared spectroscopy of the same region revealed the presence of strongly bound water at the healed crack interface, implying that chemical interaction at the fracture surface accompanied mechanical closure. This observation supports a hybrid healing mechanism: residual elastic energy and dislocation activity drive the physical re-contact of crack surfaces, while subsequent adsorption of structurally bound water may facilitate interfacial bonding or passivation. The coupling of mechanical and chemical processes offers a plausible pathway for time-dependent crack healing in geological materials under low-temperature, low-fluid conditions.

The residual stress reconfiguration parallels phenomena observed in metallic systems, where extrinsic shielding mechanisms and intrinsic dislocation processes alter crack-driving forces post-loading[2]. In a nominally brittle material like calcite, such behavior is notable and indicates an unexpectedly active post-fracture stress redistribution mechanism. Crack healing via closure driven by residual stress has been inferred in calcite at elevated temperatures[57], but the present data confirm such behavior under near-ambient conditions, where spontaneous strain reconfiguration occurred without further external stimulus. While the force exerted on the sample by the DT device during in-situ Laue diffraction could not be measured due to physical geometric constraints of the experiment, crack healing is supported by offline re-opening measurements, the data for which is to be included in another publication. Partial recovery of strength in forward-backwards-forward fracture propagation experiments has been observed in mica at humid conditions[48]. The temporal coherence of strain accumulation suggests that the crack closure process is driven not solely by mechanical unloading but also by time-dependent anelastic relaxation and dislocation recovery. This supports interpretations of spontaneous fracture healing in silicate glasses[58] and points to comparable processes in geological carbonates, governed by internal stress gradients and surface chemistry.

Traditional models of crack healing in brittle minerals like calcite often emphasize thermally activated surface diffusion or dissolution-precipitation as primary mechanisms[59,60]. While these processes undoubtedly dominate in high-temperature, fluid-rich environments, the results of this study suggest that mechanical drivers—specifically, residual elastic energy and dislocation migration—can initiate healing even under nominally dry, ambient conditions. Recent modeling efforts have emphasized the role of stress-coupled vacancy diffusion in promoting defect migration toward free surfaces[61]. These stress gradients not only drive dislocation motion but can promote local mass transfer, thereby enhancing the healing of microcracks without external chemical flux. This framework helps explain why time-dependent strain redistribution is observed even in the absence of measurable environmental changes. Additionally, surface forces may contribute to interfacial adhesion and bonding. Studies using a Surface Forces Apparatus, which measures the force of interaction between two surfaces, demonstrated that reactive calcite surfaces exhibit significant attractive forces when hydrated, which could further assist fracture closure under geologic conditions[62]. While no fluid was introduced in the current experiment, it is plausible that ambient humidity or chemisorbed species provided enough interfacial mobility to allow physical contact and limited bonding across the crack. The healing process observed here may represent the first phase of a multi-stage mechanism, beginning with mechanically driven crack closure via dislocation redistribution, followed by surface-mediated re-bonding under appropriate environmental conditions.

These findings indicate that the stress imparted by the double-torsion loading system was sufficient to mobilize dislocations along crystallographic glide planes that intersect the crack. The convergence of strongly bound water and dislocations at the free surface and their subsequent evolution suggests a stress-driven, chemically-coupled relaxation mechanism that contributes to restoring tensile strength perpendicular to the fracture. This re-establishment of compressive stress and loss of internal lattice distortion are hallmarks of spontaneous crack closure and structural recovery. These results provide direct evidence of dislocation-mediated crack healing in calcite under ambient conditions, bridging nanoscale plasticity with macroscale mechanical recovery. The application of time-resolved synchrotron microdiffraction, in combination with spectroscopic tools, opens a powerful avenue for probing the interplay of stress, plasticity, and fluid-mineral interactions in macroscopically brittle crystalline materials.

The interplay between plasticity, residual strain evolution, and environmental conditions on crack healing in calcite has far-reaching implications for geomechanical systems. In many subsurface environments, such as geothermal reservoirs, carbonate-hosted hydrocarbon plays, and fault zones, the long-term mechanical integrity and permeability of rocks are shaped by the balance between fracture generation and healing. The spontaneous closure of cracks in calcite under ambient conditions challenges prevailing assumptions that healing requires hydrothermal or chemically saturated conditions. Instead, our results support a model in which residual elastic energy and dislocation motion are sufficient to re-establish mechanical continuity. This mechanism offers a physical basis for the "healing" observed in carbonate-rich fault zones following stress perturbations[63], and could explain permeability reductions during shut-in periods in fractured reservoirs. The spontaneous healing observed here may also contribute to fault zone strength heterogeneity. Dislocation interactions can induce localized hardening and stress redistribution, influencing rupture propagation and energy dissipation during seismic events[64,65]. Incorporating time-dependent plastic healing into models of fault reactivation may refine predictions of seismic hazard in carbonate terrains.

## Methods

### Sample preparation

A rectangular slice of calcite measuring 30 mm × 10 mm × 1 mm was prepared from an optical-grade natural single crystal of Brazilian provenance (sourced and cut by Crystran, Ltd., UK). The crystallographic orientation of the slice relative to the three primary cleavage planes is shown in Fig. 1a. A pre-notch was introduced to control crack initiation and propagation direction. The sample was then mounted in a custom-designed double-torsion device to initiate crack propagation. The slice was oriented such that the induced crack would propagate along the long axis of the sample, parallel to the (10-14) cleavage plane (Fig. 1b). Crack initiation was achieved by a small ball-bearing placed at the tip of a cup-tipped set screw, which applied a controlled 3-point bending force directly over the notch. This design enabled precise control over crack growth under Mode I loading conditions. The double-torsion device was modified to hold the sample perpendicular to the X-ray beam (Supplementary Fig. 1).

### Laue microdiffraction

The experiment was conducted at ALS beamline 12.3.2 using a poly-chromatic beam with a spot size of 1 μm and an energy range of 5–24 keV. The beamline was configured in transmission geometry and calibrated using a YAG ($Y_3Al_5O_{12}$) crystal mounted on a glass substrate. A 90° angle of incidence was used to uniformly sample the crack plane across the scanned area. After defining the 2D scan area with beamline control software, the stage advanced in discrete steps, collecting one Laue diffraction image at each position, using a DECTRIS Pilatus 1 M placed directly above the sample, with an exposure time of 1 s per frame.

Prior to crack propagation, a control scan (1.5 mm × 2 mm) was misplaced and collected slightly above the anticipated crack plane due to the inability to predict the crack propagation path. This control scan

was used to establish a baseline strain field (control, Fig. 2). The sample was then manually loaded using the ball-bearing-tipped actuator, propagating a fresh tensile crack ~15 mm in length along the (10-14) plane. Permanent ink fiducial markers were applied to the sample surface to mark the location of the crack tip while under load. The load was then gradually reduced. Upon partial unloading, spontaneous crack closure was observed: the crack tip retreated left toward the ball bearing and the area that previously displayed the crack while under load became optically indistinguishable from the surrounding crystal, indicating a self-healing response. To investigate the evolution of strain during this healing process, three high-resolution scans were collected in the area of the former crack tip (cracktip_unloaded 1, 2, 3; Fig. 2). Each scan measured ~0.17 mm × 1.4 mm, with a reduced vertical range in cracktip_unloaded 3 (shortened by 50 µm) to accommodate time constraints. The approximate location of the former crack tip relative to the cracktip_unloaded scans 1, 2, and 3 is visualized in Fig. 2.

Step sizes, or the distance of stage translation between collected diffraction images, varied depending on the scan purpose. The control scan used large step sizes (10 µm) to cover a broader region at lower resolution. In contrast, the cracktip_unloaded scans used fine step spacing (2 µm) to generate high-resolution strain maps of the former crack tip area. Due to the small step size and extended scan area, these datasets were time-intensive: cracktip_unloaded scans 1 and 2 each collected ~59,000 diffraction images over ~16 hours, and scan 3 collected ~45,000 patterns in 12 hours. As such, each scan should be considered a temporally distributed dataset, with several hours elapsing between the first and last diffraction image collected within each map. Strain for each diffraction pattern was calculated using XMAS software.

## Infrared spectroscopy

The infrared hyperspectral imaging was performed using a Fourier Transform Infrared (FTIR) microscope (Cary 620 FTIR microscope and Cary 670 Spectrometer, Agilent Technologies) equipped with a liquid-nitrogen-cooled focal-plane array (FPA) detector at ALS beamline 2.4. This configuration enables acquisition of spatially resolved spectral data with a diffraction-limited spatial resolution of approximately 4 µm under a ×25 objective (N.A. = 0.81), providing fine-scale chemical imaging of the sample. Spectra were collected by co-adding 100 individual scans to enhance the signal-to-noise ratio, covering the mid-infrared spectral range from 700 cm$^{-1}$ to 4000 cm$^{-1}$. This range encompasses key vibrational modes, including the carbonate symmetric stretching mode characteristic of calcite (~1450 cm$^{-1}$), the O-H bending mode of molecular water (~1620 cm$^{-1}$), and the broad O-H stretching band (~3200 cm$^{-1}$). These spectral features enable simultaneous mapping of mineralogical and fluid phases within the sample. The spatial distribution of water was reconstructed by integrating the absorption intensity of the O-H bending mode at ~1620 cm$^{-1}$, which provides a quantitative proxy for local water content based on the Beer-Lambert law.

## Dislocation analysis

A dislocation analysis was performed using PYXIS software[39,66] to evaluate activity along four common slip systems in calcite: {10-14}[1-210], r-slip{10-14}[-2012], and f-slip (01-12)[0-111] and (01-12)[-2021]. The analysis compared slip system activity over the duration of the experiment. The diffraction image was collected at the same stage coordinates for the three unloaded scans, taken from the center of the crack plane. Only peaks with complete and well-defined intensity profiles were considered; any partial or truncated peaks were excluded. Peaks for which a satisfactory fit could not be obtained were discarded from further analysis. To ensure robust identification of slip system activation, only peaks with angular deviations of less than 10° between the measured elongation axis and the predicted slip direction were included. This filtering step helped isolate those peaks most reliably associated with specific crystallographic slip systems.

## Data availability

The data generated in this study are provided in the Source Data file. Due to the large file size, diffraction images can be made available from the authors upon request. Source data are provided with this paper.

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

## Acknowledgements

The authors would like to thank Kai Chen and Jiawei Kou for their assistance with PYXIS. This study was part of the PhD thesis of M.C.D. at UC Berkeley. This research used beamlines 12.3.2 and 2.4 of the

ALS, which is a DOE Office of Science User Facility under contract no. DE-AC02-05CH11231. M.C.D. was supported in part by an ALS Doctoral Fellowship in Residence as well as the U.S. Department of Energy, Office of Science, Office of Workforce Development for Teachers and Scientists, Office of Science Graduate Student Research (SCGSR) program. The SCGSR program is administered by the Oak Ridge Institute for Science and Education (ORISE) for the DOE. ORISE is managed by ORAU under contract number DE-SC0014664. All opinions expressed in this paper are the author's and do not necessarily reflect the policies and views of DOE, ORAU, or ORISE. H.P.L., S.N., and Z.H. were supported by the U.S. Department of Energy, Office of Science, Office of Basic Energy Sciences, Chemical Sciences, Geosciences, and Biosciences Division, through its Geoscience program at LBNL under Contract DEAC02-05CH11231. M.C.D. would also like to acknowledge support from the U.S. Department of Energy, Office of Science Energy Earth-shot™ Initiative, as part of the "Center for Coupled Chemo-Mechanics of Cementitious Composites for EGS (C4M)" project at Brookhaven National Laboratory under contract number 2026-BNL-IS012-FUND, and by the Geothermal Technologies Office in the US Department of Energy (DOE) Office of Energy Efficiency and Renewable Energy (EERE), under the auspices of the US DOE, Washington, DC, USA, under contract no. DE-AC02-98CH 10886. H.R.W. is appreciative of support from DOE-BES (DE-FG02-05ER15637) and NSF (EAR-2154351).

## Author contributions

Conceptualization: M.D., H.P.L., S.N. Methodology: N.T., M.D., H.P.L., Z.H. Investigation: M.D., H.P.L., Z.H. Visualization: M.D., N.T. Supervision: H.R.W. Writing—original draft: M.D., H.P.L. Writing—review & editing: H.R.W., N.T., S.N., Z.H., H.P.L., M.D.

## Competing interests

The authors declare no competing interests.
