## [Transparent Peer Review file · Nature Communications]

Spontaneous crack healing in calcite reveals the influence of dynamic strain evolution and surface chemistry

Corresponding Author: Dr Michelle Devoe

Version 0:

Reviewer comments:

Reviewer #1

(Remarks to the Author)
Dear editor,

I have reviewed the manuscript titled "Spontaneous Crack Healing in Calcite: Insights from Synchrotron Laue Microdiffraction and Infrared Spectroscopy." This study presents a detailed investigation into the mechanisms of subcritical crack growth and healing in calcite, using advanced synchrotron-based techniques and infrared spectroscopy. The authors document the progressive intensification of compressive strain perpendicular to the crack plane (ϵ_{yy}) and the accumulation of tensile strain parallel to the crack (ϵ_{zz}) over a 44-hour period. This provides direct evidence of mechanically driven crack closure under ambient conditions. The results are noteworthy as they challenge the traditional view that crack healing in brittle minerals like calcite requires elevated temperatures or fluid saturation.

The article is very well written, providing a clear flow between the explanation of the methodology (experimental design, novelty, complexity), the results and their impact. This work is worthy of publication in Nature Communications because:

- this work shed light on the mechanisms of spontaneous crack healing (SCH) under ambient conditions by demonstrating that dislocation-mediated plasticity and water accumulation can drive fracture closure without external stimuli in calcite. The results align with recent studies on dislocation recovery in calcite and extend these findings to ambient conditions, which is a significant advancement;
- the study offers a physical basis for understanding permeability reductions during shut-in periods and the mechanical stability of carbonate-rich fault zones;
- the findings provide critical insights into permeability evolution in geothermal and hydrocarbon reservoirs, where fracture healing can significantly impact fluid transport and reservoir performance.

Results and conclusions are well-supported by the data, and reproducibility is possible.

Upon reviewing this article, I only noted five points for clarification for me.

- 1) A main reference (e.g.) to Laue (micro)diffraction is missing.
- 2) In Table 1, it is unclear what the standard deviations are for: min, max, non presented average Von Mises values? At the moment, one could read that the min values could be negative which is not possible by definition.
- 3) One area that could benefit from further clarification is the interpretation of water distribution. The authors suggest that water accumulation may result from diffusion or adsorption along microcracks. While the data support the presence of water, the exact mechanism (diffusion vs. adsorption) could be explored further, possibly through additional modeling or experimental validation.
- 4) As each scan took several hours to complete, how did the temporal artifacts were accounted for in the analysis?
- 5) Do certain slip systems dominate the process, or is their activity uniformly distributed?

Thank you for the opportunity to review this manuscript.

Reviewer #2

(Remarks to the Author)
Dear authors, dear editor,

The manuscript entitled “Spontaneous crack healing in calcite revealed: Strain evolution, dislocation dynamics and surface chemistry” by Michelle Devoe and co-workers presents results from high resolution mapping of the strain field surrounding a healing crack in a calcite crystal. Using state of the art synchrotron technology, the authors demonstrate that cracks in calcite can heal significantly at nominally ambient conditions driven by a combination of dislocation motion and water-mediated chemical interactions.

The manuscript is well written and engaging. The methods are at the forefront of the field, and the results challenge the prevailing view that crack healing in calcite requires elevated temperature or pressure. My only reservation concerns the introduction, which I found somewhat limited in its review of the current state of knowledge on crack healing (see my line-by-line comments). I recommend publication after minor revisions.

Best regards,
Gabriel Meyer

Line by line comments:

L52: Here, I must kindly disagree with the use of “brittle geologic materials” in this sentence. In the rock mechanics literature, “brittle” refers to materials in which no crystal plasticity occurs. A comprehensive description of the nomenclature generally followed in the field can be found in Rutter et al., 1986.

L56: I suggest adding Fredrich et al., 1989 to the references, as it is widely regarded as a seminal study on marble deformation.

L60: Could you give a reference for a geo-reservoir hosted in calcitic rocks here? I only see those for cements and faults.

L61: What is meant by fault healing here? The cited reference studies calcite sealing which I believe is quite different from the process that is the subject of the manuscript.

L71: The connection to the opening of this paragraph on crack healing and sealing is not sufficiently developed. Where does spontaneous crack healing fit within the broader range of crack-healing processes? How does it differ from sealing? Under what conditions can SCH be expected? Further clarification would strengthen this section.

L73: From reading the abstract, citation 19 does not appear to support the point being made here as it refers to SCG and not SCH. For SCH impact on mechanical and seismic properties in calcitic rocks, I would suggest Schubnel et al., 2005 or Meyer et al., 2021. The latter, in particular, shows crack healing in calcite at ambient temperature and humidity, which might be of interest to you.

L89: Overall, the introduction does not provide a sufficient state-of-the-art overview of crack self-healing. Several seminal studies already exist and should be acknowledged, including Smith et al. (1984), Evans et al. (1977), and Hickman et al. (1987).

L117: Is there a reason why the control scan is not acquired where the crack will eventually propagate?

L126: I believe the scan map in supplementary should be part of the main text. I personally found hard to visualize the position of the different scans from the text only.

129: Maybe specify “displacement steps” for clarity.

L212: First reference to “north” in text. I would suggest homogenizing. I believe earlier the control position was referred to as “above” the crack.

L261: Where is this water originally from? Is it just room humidity?

L276: The shape of the water halo as well as that of the deviatoric strain on figure 5 appears asymmetrical — being wider above the healed crack. Is this an artefact? Could you comment?

L277-282: Citation needed to support these statements.

References:

Evans, A. G. and Charles, E. A. (1977). Strength recovery by diffusive crack healing. *Acta Metallurgica*, 25(8):919–927.

Fredrich, J. T., Evans, B., & Wong, T. F. (1989). Micromechanics of the brittle to plastic transition in Carrara marble. *Journal of Geophysical Research: Solid Earth*, 94(B4), 4129-4145.

Hickman, S. H. and Evans, B. (1987). Influence of geometry upon crack healing rate in calcite. *Physics and Chemistry of Minerals*, 15(1):91–102.

Meyer, G. G., Brantut, N., Mitchell, T. M., Meredith, P. G., and Plumper, O. (2021). Time dependent mechanical crack closure as a potential rapid source of post-seismic wave speed recovery: Insights from experiments in Carrara Marble. *Journal of Geophysical Research: Solid Earth*, 126(4):e2020JB021301.

Rutter, E. H. (1986). On the nomenclature of mode of failure transitions in rocks. *Tectonophysics*, 122(3-4), 381-387.

Schubnel, A., Fortin, J., Burlini, L., and Gueguen, Y. (2005). Damage and recovery of calcite rocks deformed in the cataclastic regime. *Geological Society, London, Special Publications*, 245(1):203–221.

Smith, D. L. and Evans, B. (1984). Diffusional crack healing in quartz. *Journal of Geophysical Research: Solid Earth*, 89(B6):4125–4135.351

Reviewer #3

(Remarks to the Author)

Review of 634938_0 submitted to Nature Communications:

Summary:

The manuscript entitled “Spontaneous crack healing in calcite revealed: Strain evolution, dislocation dynamics and surface chemistry” by Michelle Devoe, Harrison Lisabeth, Seiji Nakagawa, Zhao Hao, Nobumichi Tamura, and Hans-Rudolf Wenk presents experimental measurements of crack healing based on time-resolved synchrotron Laue X-ray diffraction, optical measurements of crack opening, and scanning infrared spectroscopy. The main result that negative axial strains develop near the crack surface upon unloading, accompanied by increased plastic activity and localized water migration, suggesting a chemically-driven healing mechanism.

Evaluation:

The study is original and supported by a rich experimental dataset. It has significant potential to advance our understanding of fracture and healing processes in reservoir rocks. The manuscript is generally well written. However, the definition of the crack tip (a key concept for interpreting the measurements) is ambiguous and leads to confusion. In addition, while crack healing is mentioned, it is not (mechanically) assessed. I recommend publication in Nature Communications after major revisions

The authors will find below detailed comments on the manuscript.

Detailed comments:

- Introduction – The introduction is well-written. The motivation and broader relevance of the study are clearly stated. The results are likely to interest a wide audience across geomechanics, geophysics, and (to some extent) mechanical and civil engineering.

- Section 2 “Results”

Remark 2.1: The dimensions of the sample could be added to the main text, even if they are reported in the methods section.

Remark 2.2: I expect that the diffraction pattern obtained from transmission Laue measurements represents an average response through the sample thickness (if the energy is high enough), whereas both the optical observations of crack opening and the infrared spectroscopy data correspond to surface measurements. However, during a double-torsion test, the crack front is curved, as shown by the authors in Figure 1. Is the curvature negligible with respect to the dimension of scanned area? If it is not the case, how can the authors reconcile bulk-averaged (Laue) and surface (optical and IR) measurements? Specify whether the surface measurements are performed on the side of shorter or longer crack length?

Remark 2.3: The notion of the crack tip is not clearly defined throughout the manuscript. At line 125, the authors refer to “the former crack tip after unloading”, which I understand as the right end of the optically visible crack opening prior to unloading. Later, at line 131, they mention “high-resolution strain maps around the healed crack tip”. It is unclear whether this refers to the same position as the “former crack tip” or to a distinct location. In Figure 2a/b/c, the “crack plane” is indicated by a white line, but since the line has finite length, its right edge could also be interpreted as a (former/new/healed?) crack tip. Figure 5 shows a visible and a healed crack. I recommend clarifying this definition early in the manuscript and keep the same terminology.

Remark 2.4: Assuming plane stress conditions, I would expect ϵ_{zz} to be a fraction of ϵ_{yy} (assuming negligible ϵ_{xx}), i.e. $|\epsilon_{zz}| < |\epsilon_{yy}|$. The opposite signs are correct. Yet, $|\epsilon_{zz}| > |\epsilon_{yy}|$. Can the authors comment on that? Is it an artifact introduced by subtracting the baseline strain?

Remark 2.5: In Figures 3a and 4a, the caption mentions diffraction peaks boxed in red and white, but there is no such thing in the figure.

Remark 2.6: The IR spectroscopy measurements are performed on the surface of the specimen. Do the authors expect the same signal near the crack surface within the material (in the z direction)?

- Section 3 “Discussion”

Remark 3.1: If the crack indeed healed, the force required to propagate upon reloading should exceed that previously needed to reach the initial crack length before unloading. I understand that in situ force measurements were not possible during the synchrotron experiments due to technical constraints. However, do the authors have any supporting evidence for crack healing from complementary tests conducted outside the synchrotron, where the setup was instrumented with a force

sensor?

Remark 3.2: In Figure 2, the region of axial strain goes behind the white line, which I identify as the visible crack tip after unloading. Does this mean that the crack healed in an opened region? Did the authors observe a decrease of crack opening in that region, as it would be expected during healing? I do not know how to interpret this without a clear answer to my previous remark 2.2

Remark 3.3: Traditional healing cohesive-zone models include a minimum opening, below which healing process are activated. Do these measurements provide way to validate such models?

Minor comments:

- the notations SCG and SCH of line 67 and 71 are already defined at line 47.
- in figure 3, numbering of the subfigures are in capital letters instead of lowercase.
- at line 290 and 291, the symbol epsilon is not consistent with that used in section 2 and Figures 2 and 5.

Version 1:

Reviewer comments:

Reviewer #2

(Remarks to the Author)

Dear Editor, dear Authors,

This is the second time I have reviewed the manuscript entitled "Spontaneous crack healing in calcite reveals the influence of dynamic strain evolution and surface chemistry", by Michelle Devoe, Harrison Lisabeth, Seiji Nakagawa, Zhao Hao, Nobumichi Tamura and Hans-Rudolph Wenk.

In my initial review, most of my concerns related to the introduction and the supporting literature. I am happy to report that these points have now been satisfactorily addressed.

I believe the manuscript is now suitable for publication in Nature Communications.

Best regards,
Gabriel Meyer

Reviewer #3

(Remarks to the Author)

The authors have carefully addressed all points raised in my first review. The notion of crack tip is clearer, and makes the manuscript easier to read. The experiments are carefully conducted, and were cleverly designed to reconcile surface and depth-averaged multi-field measurements, which is no small feat. I am therefore happy to recommend the manuscript for publication.

Please note that the line references the Authors provide in this document correspond to the Track Changes “All Markup Visible” version of the Word document.

RESPONSE TO REVIEWER COMMENTS

Reviewer #1 (Remarks to the Author):

Upon reviewing this article, I only noted five points for clarification for me.

1) A main reference (e.g.) to Laue (micro)diffraction is missing.

References to previous studies that used Laue diffraction to study the elastic strain in bulk samples have been added: Spolenak et al. (2016), Devoe et al. (2023), and Chen et al. (2016) (lines 92-95). References on how elastic strain is measured from Laue diffraction patterns have been added: Tamura (2014), Wenk et al. (2020), Devoe et al. (2023) (line 174-175).

2) In Table 1, it is unclear what the standard deviations are for: min, max, non presented average Von Mises values? At the moment, one could read that the min values could be negative which is not possible by definition.

Average values have been added to Table 1 (line 217). The Table 1 caption has also been modified to clarify that the strain values presented in the table are the corrected values. A few strain values in the table erroneously reported the uncorrected data values and have been replaced with the corrected strain values, and the text was also modified to reflect these changes.

3) One area that could benefit from further clarification is the interpretation of water distribution. The authors suggest that water accumulation may result from diffusion or adsorption along microcracks. While the data support the presence of water, the exact mechanism (diffusion vs. adsorption) could be explored further, possibly through additional modeling or experimental validation.

We agree additional experimental work is needed to identify the mechanism. A response addressing this and the reasoning for our presumed mechanism has been added to lines 352-359.

4) As each scan took several hours to complete, how did the temporal artifacts were accounted for in the analysis?

Scans were not corrected for temporal artifacts as strain rates could not be quantified and time-dependent slip system activities could not be resolved. Clarifying language addressing this has been added to lines 170-172, and lines 330-333.

5) Do certain slip systems dominate the process, or is their activity uniformly distributed?

In regards to identifying the dominant deformation process present in the experiment, only a small subset of possible slip systems was studied on a limited number of peaks on one diffraction pattern due to time constraints, so no dominant slip system to describe the deformation process could be determined with certainty. In regards to the PYXIS analysis performed, neither slip system (r or f) was significantly more active than the other, so out of the four slip systems investigated, neither could be determined as more dominant. In regards to spatial distribution, only one diffraction pattern was analyzed for this experiment due to the amount of time required to perform the analysis, so no differences in slip system activity across the scan area were determined. Additionally, PYXIS analyses on more diffraction patterns across the scan area could reveal more insights into the dominant deformation processes and their spatial distribution. Sentences addressing this have been added to lines 275-286.

Reviewer #2 (Remarks to the Author):

L52: Here, I must kindly disagree with the use of “brittle geologic materials” in this sentence. In the rock mechanics literature, “brittle” refers to materials in which no crystal plasticity occurs. A comprehensive description of the nomenclature generally followed in the field can be found in Rutter et al., 1986.

The Authors thank the Reviewer for raising this point. We, however, diverge and suggest crystal plasticity can occur even in materials that fail in a macroscopically brittle manner. In rock mechanics, the terms “brittle” and “ductile” are typically used to distinguish macroscopic post-failure behavior, with brittle materials showing strain softening and ductile materials exhibiting strain-neutral or strain-hardening behavior prior to softening. These classifications do not imply the absence of micro-scale plasticity, and rocks commonly express mixed micromechanical processes that vary with pressure, temperature, and chemistry. Our original use of the term “brittle” was intended to contrast calcite-rich geomaterials with metals, whose mechanical response at ambient conditions is dominated by crystal plasticity. Calcite-bearing rocks at low pressure and temperature are generally understood to behave macroscopically brittle, despite accommodating some crystal plasticity at the grain scale. To avoid any ambiguity, we have revised the text to use the term “macroscopically brittle” in place of “brittle,” and an explanation which more precisely conveys the intended meaning and avoids implying the absence of plasticity (lines 53-54, 507).

L56: I suggest adding Fredrich et al., 1989 to the references, as it is widely regarded as a seminal study on marble deformation.

A reference to Fredrich et al. (1989) has been added to the statement of historic calcite deformation studies (line 58).

L60: Could you give a reference for a geo-reservoir hosted in calcitic rocks here? I only see those for cements and faults.

A reference to the Ekofisk Field, a chalk hydrocarbon reservoir, has been added (Agarwal et al. 2000), and a reference to the common occurrence of calcite, deposited as scale within a geothermal reservoir, in this case Miravalles in Costa Rica, has been added (Vaca et al. 1989) (line 62).

L61: What is meant by fault healing here? The cited reference studies calcite sealing which I believe is quite different from the process that is the subject of the manuscript.

The Authors thank the Reviewer for this comment, however cannot find mention of fault healing at line 61 of the original, submitted manuscript or in the *Introduction* section. Could it be possible that this comment is in reference to the mention of “fault zone stability” (line 79), with references to Carpenter et al. (2025), Olsen et al. (1993), and Orlecka-Sikora & Cielesta (2020)? These references refer to the deformation response of calcitic-rocks under shear stress, subcritical crack growth, and subcritical rupture processes, which are integral to fault zone mechanical stability. Indeed, crack sealing is much different from spontaneous crack healing. An explanation of the differences and supporting references have been added and are addressed in the following comment/response.

L71: The connection to the opening of this paragraph on crack healing and sealing is not sufficiently developed. Where does spontaneous crack healing fit within the broader range of crack-healing processes? How does it differ from sealing? Under what conditions can SCH be expected? Further clarification would strengthen this section.

An explanation of crack sealing (lines 68-71) and how it differs from crack healing (lines 71-76) has been added. Spontaneous crack healing can occur upon removal of the driving force of crack propagation, and this clarifying point has been added (line 81-82).

L73. From reading the abstract, citation 19 does not appear to support the point being made here as it refers to SCG and not SCH. For SCH impact on mechanical and seismic properties in calcitic rocks, I would suggest. The latter, in particular, shows crack healing in calcite at ambient temperature and humidity, which might be of interest to you.

The Authors thank the Reviewer for contributing these supportive citations demonstrating the influence of crack healing on seismic wave velocities. The citation to the reference previously numbered #19 (Orlecka-Sikora & Cielesta, 2020) has been removed and replaced with citations to Schnubel et al. 2005 and Meyer et al. 2021 (line 83).

L89: Overall, the introduction does not provide a sufficient state-of-the-art overview of crack self-healing. Several seminal studies already exist and should be acknowledged, including Smith et al. (1984), Evans et al. (1977), and Hickman et al. (1987).

Additional background information and references to the seminal studies on crack healing and mechanisms responsible have been added (lines 68-80).

L117: Is there a reason why the control scan is not acquired where the crack will eventually propagate?

A sentence has been added clarifying that the control scan coordinates were intended to but ultimately did not overlap with the scan area due to the Authors' inability to predict crack propagation path (lines 140-144; lines 573-574). Language has been added to clarify the purpose of the control scan (lines 140-142).

L126: I believe the scan map in supplementary should be part of the main text. I personally found hard to visualize the position of the different scans from the text only.

Supplementary Figure 2 from the original manuscript, showing the relative scan positions, has been moved from Supplementary Materials into the main text as the new Figure 2 (line 133) and the subsequent figures in the Manuscript and Supplementary Materials have been renumbered accordingly.

L29: Maybe specify “displacement steps” for clarity.

Clarifying language that “step size” refers to the length of stage displacement has been added to lines 116-118 and line 588.

L212: First reference to “north” in text. I would suggest homogenizing. I believe earlier the control position was referred to as “above” the crack.

“North” has been changed to “above” (line 266).

L261: Where is this water originally from? Is it just room humidity?

The presence of water is attributed to ambient humidity. A clarifying sentence has been added (lines 316-317).

L276: The shape of the water halo as well as that of the deviatoric strain on figure 5 appears asymmetrical — being wider above the healed crack. Is this an artefact? Could you comment?

Indeed, the asymmetry of the halo above and below the crack is an intriguing feature. Due to the temporal distortion of the scan, with ~14-16 hours difference between the collection of the first and last diffraction images of each scan, it is possible that this asymmetry is a temporal artefact and therefore the Authors refrain from overinterpretation of this feature. A clarifying sentence has been added (lines 330-333).

L277-282: Citation needed to support these statements.

Because this paragraph, which provided discussion on the interpretation of the water signal and its mechanism of placement, is redundant with the paragraph that followed, this paragraph has been deleted (lines 341-348), and language on the explanation of the interpretation of the water signal and possible water emplacement mechanisms, and supporting references, have been added to lines 337-340, and lines 349-359.

References:

Evans, A. G. and Charles, E. A. (1977). Strength recovery by diffusive crack healing. *Acta Metallurgica*, 25(8):919–927.

Fredrich, J. T., Evans, B., & Wong, T. F. (1989). Micromechanics of the brittle to plastic transition in Carrara marble. *Journal of Geophysical Research: Solid Earth*, 94(B4), 4129-4145.

Hickman, S. H. and Evans, B. (1987). Influence of geometry upon crack healing rate in calcite. *Physics and Chemistry of Minerals*, 15(1):91–102.

Meyer, G. G., Brantut, N., Mitchell, T. M., Meredith, P. G., and Plumper, O. (2021). Time dependent mechanical crack closure as a potential rapid source of post-seismic wave speed recovery: Insights from experiments in Carrara Marble. *Journal of Geophysical Research: Solid Earth*, 126(4):e2020JB021301.

Rutter, E. H. (1986). On the nomenclature of mode of failure transitions in rocks. *Tectonophysics*, 122(3-4), 381-387.

Schubnel, A., Fortin, J., Burlini, L., and Gueguen, Y. (2005). Damage and recovery of calcite rocks deformed in the cataclastic regime. *Geological Society, London, Special Publications*, 245(1):203–221.

Smith, D. L. and Evans, B. (1984). Diffusional crack healing in quartz. *Journal of Geophysical Research: Solid Earth*, 89(B6):4125–4135.351

Reviewer #3 (Remarks to the Author):

Evaluation:

The study is original and supported by a rich experimental dataset. It has significant potential to advance our understanding of fracture and healing processes in reservoir rocks. The manuscript is generally well written. However, the definition of the crack tip (a key concept for interpreting the measurements) is ambiguous and leads to confusion. In addition, while crack healing is mentioned, it is not (mechanically) assessed. I recommend publication in *Nature Communications* after major revisions

The authors will find below detailed comments on the manuscript.

Detailed comments:

• Introduction – The introduction is well-written. The motivation and broader relevance of the study are clearly stated. The results are likely to interest a wide audience across geomechanics, geophysics, and (to some extent) mechanical and civil engineering.

The Authors thank the Reviewer for this feedback.

• Section 2 “Results”

Remark 2.1: The dimensions of the sample could be added to the main text, even if they are reported in the methods section.

The dimensions of the sample have been added to section 2.1 (line 108).

Remark 2.2: I expect that the diffraction pattern obtained from transmission Laue measurements represents an average response through the sample thickness (if the energy is high enough), whereas both the optical observations of crack opening and the infrared spectroscopy data correspond to surface measurements. However, during a double-torsion test, the crack front is curved, as shown by the authors in Figure 1. Is the curvature negligible with respect to the dimension of scanned area? If it is not the case, how can the authors reconcile bulk-averaged (Laue) and surface (optical and IR) measurements? Specify whether the surface measurements are performed on the side of shorter or longer crack length?

As the Reviewer points out, Laue X-ray diffraction measurements, optical, and IR measurements do not examine the same sample volume in our experiment; Laue X-ray diffraction probes the entire thickness of

the sample due to the high energy X-ray beam and produces a bulk-averaged strain measurement, whereas infrared (IR) measurements probe only several micrometers into the surface of the sample. Optical measurements were limited to crack position and optically-observed healing. Both optical imaging and IR imaging were conducted on the tensile side (long-crack side) of the sample. Additional language clarifying the surficial-nature of the IR measurements has been added to the manuscript (lines 327-330). The curved crack front of a double torsion experiment produces a depth-dependent strain field, and while depth-resolved Laue diffraction has been done at other synchrotrons, the technique is not available at Advanced Light Source beamline 12.3.2. Although we currently do not have an established method for evaluating and correcting for this discrepancy, we will describe below how the “uncorrected” strain data should be understood and considered when compared to other surface-based measurements. In the Supplementary Materials section, we added the new Supplementary Figure 3 as a supporting visual, which is also provided below.

In our experiment, the incident X-ray beam contains high-energy wavelengths (>20 keV) that can penetrate through the entire sample thickness. Therefore, diffraction can originate at any point along the X-ray beam path through the sample thickness. For this reason, each Laue diffraction image contains the bulk-averaged signal of the sample at that sample stage coordinate position (i.e. 2D positions within the scan area). Double torsion experiments typically produce a curved crack front that intersects the tensile side of the sample surface at a high angle. The crack front rapidly changes slope on the order of the sample thickness, then asymptotically becomes parallel to the sample surface (e.g. Nakagawa et al. 2023). The highly concentrated stress around the crack front creates a localized damage zone (“process zone”) containing microcracks and plastic deformation including dislocations. Because plastic deformation is generated mostly within the narrow process zone along the crack front, once the crack front moves on and the crack is fully propagated, we can think that plastic deformation is distributed evenly along the crack surface at any given sample depth (or crack height). However, unfortunately, the rotation of the strain field along the curved crack front does not allow us to have a simple relationship between the crack height (along a beam path) and the averaging effect.

Because the strain measured by Laue X-ray diffraction is a measure of average strain along the beam path through the thickness of the sample, the measured strain can vary depending on the location of the X-ray beam path relative to the crack front geometry. In our experiment, conducted after crack healing, the crack front retreated following unloading and remained outside of the X-ray diffraction scan area. Supplementary Figure 3c shows how Path A intersects a larger volume of distorted lattice than Path B; and therefore, diffraction patterns collected along Path A will have greater strain than those collected along Path B. This variation in strain magnitude can be seen most clearly in the ϵ_{yy} strain tensor component (Fig. 3a); strain magnitude increases to the left, towards the notch. In lines 384-396, we provided an explanation on the cumulative nature of Laue diffraction patterns, the asymmetry of the crack front, and the difference in strain magnitude due to the curvature of the crack front, which we have discussed in the above.

Remark 2.3: The notion of the crack tip is not clearly defined throughout the manuscript. At line 125, the authors refer to “the former crack tip after unloading”, which I understand as the right end of the optically visible crack opening prior to unloading. Later, at line 131, they mention “high-resolution strain maps around the healed crack tip”. It is unclear whether this refers to the same position as the “former crack tip” or to a distinct location. In Figure 2a/b/c, the “crack plane” is indicated by a white line, but since the line has finite length, its right edge could also be interpreted as a (former/new/healed?) crack tip. Figure 5 shows a visible and a healed crack. I recommend clarifying this definition early in the manuscript and keep the same terminology.

The Authors agree that the language used to describe the location of the scans is not consistent and can cause confusion. A definition of the “crack tip” has been added to line 146.

Following partial unloading, the crack tip retreated toward the notch, and was located outside and to the left of cracktip_unloaded scans 1, 2, and 3. Language clarifying the crack tip location while the sample was loaded (lines 147-148), following unloading and during the cracktip_unloaded scans has been added (lines 150-152, 154-155, 159-161, 580-581, 586-587).

The coordinates of scans 1, 2, and 3 surround the crack tip location while the sample was under load, as tracked using permanent ink fiducial markers (added lines 146-147, 578-579), but which became optically indistinguishable from the bulk sample following partial unloading. Because the data was collected following partial unloading of the sample and the retreat of the crack tip toward to notch-end side of the sample, clarifying language has been added to references to the “crack tip” to aid in the reader’s understanding, such as “former location” and/or “prior to unloading” (lines 151-152, 154-155, 166, 202-203, 216-217, 223, 234, 325-326, 375-376, 591).

The figure caption for Figure 3 (Figure 2 in the original manuscript) has been modified to clarify that the white line overlay in the maps for scans 1, 2, and 3 is a visual aid and its length is arbitrary (lines 230-231).

The text in Figure 6 (Figure 5 in the original manuscript) and its figure caption have been modified to clarify crack location terminology (lines 441-447).

Remark 2.4: Assuming plane stress conditions, I would expect ϵ_{zz} to be a fraction of ϵ_{yy} (assuming negligible ϵ_{xx}), i.e. $|\epsilon_{zz}| < |\epsilon_{yy}|$. The opposite signs are correct. Yet, $|\epsilon_{zz}| > |\epsilon_{yy}|$. Can the authors comment on that? Is it an artifact introduced by subtracting the baseline strain?
Laue diffraction cannot calculate the full strain tensor (dilatational strain tensor + deviatoric strain tensor) as absolute unit cell lengths cannot be determined when a white X-ray beam is used. For this reason, only the deviatoric strain tensor can be calculated which describes the relative change in shape of the unit cell, and as such, the signage convention (whether positive or negative) is arbitrary. In this study, the Authors have assigned positive values to strain which demonstrates an extension, and negative to strain values that demonstrate a contraction as compared to an ideal, unstrained calcite unit cell. Language clarifying this has been added to lines 178-181. Additional language on the calculation of strain was also added for clarification (lines 181-183).

Remark 2.5: In Figures 3a and 4a, the caption mentions diffraction peaks boxed in red and white, but there is no such thing in the figure.

References to red and white boxes in captions for Figures 3a (now 4a) and 4a (now 5a) have been replaced with references to yellow boxes to reflect what is displayed in the figure (lines 290-291, 307-308).

Remark 2.6: The IR spectroscopy measurements are performed on the surface of the specimen. Do the authors expect the same signal near the crack surface within the material (in the z direction)?
As mentioned by the Reviewer, IR measurements only probe the surface of the specimen. However, the mechanism for water attachment is likely via adsorption along microcracks and defect structures originating from the surfaces of the crack. Because these microcracks are formed due to the movement of the process zone through the material during crack propagation, they are expected to be distributed throughout the thickness of the material following crack front geometry (Supplementary Figure 2). Thus, water distribution would be expected to be distributed through the thickness of the sample following the resultant microcrack network so long as the microcracks are connected to the crack-air interface. Lines addressing this comment have been added (lines 352-359).

• Section 3 “Discussion”

Remark 3.1: If the crack indeed healed, the force required to propagate upon reloading should exceed that previously needed to reach the initial crack length before unloading. I understand that in situ force measurements were not possible during the synchrotron experiments due to technical constraints. However, do the authors have any supporting evidence for crack healing from complementary tests conducted outside the synchrotron, where the setup was instrumented with a force sensor?

Yes, we have observed strength recovery in calcite slides in the double-torsion geometry using offline tests, however the results of such are being used in another manuscript currently in preparation for submission and therefore cannot be included here. An explanation about this offline data and a citation to work done by Wan et al. 1990 on successful healing and partial recovery of strength in mica has been added to the manuscript (lines 430-435).

Remark 3.2: In Figure 2, the region of axial strain goes behind the white line, which I identify as the visible crack tip after unloading. Does this mean that the crack healed in an opened region? Did the authors observe a decrease of crack opening in that region, as it would be expected during healing? I do not know how to interpret this without a clear answer to my previous remark 2.2

The white line in Figure 2 (current Figure 3) is just a visual aid. The location of the crack tip after unloading is outside of the scan dimensions. Clarifying language on the crack tip location during the scans, and the purpose of the white lines in Figure 3 (Figure 2 in the original manuscript) have been added to lines 155-156, and lines 230-231, respectively.

Remark 3.3: Traditional healing cohesive-zone models include a minimum opening, below which healing process are activated. Do these measurements provide way to validate such models?

The Authors thank the Reviewer for this thought-provoking question. The microdiffraction measurements presented here do not directly resolve crack opening displacements at the scale required to define a precise healing threshold in terms of separation distance. However, they do provide indirect but physically relevant constraints on such models. Specifically, the measurements resolve spatial and temporal evolution of elastic strain, lattice rotation, and stress redistribution associated with crack closure and interfacial contact during healing. These observables reflect the re-establishment of load-bearing pathways across previously damaged regions, even when absolute crack apertures cannot be measured directly. In this sense, our results can be viewed as constraining the mechanical consequences of healing, namely, the recovery of stiffness and stress transmission, rather than the geometric criterion (minimum opening) itself.

Minor comments:

• **the notations SCG and SCH of line 67 and 71 are already defined at line 47.**

Duplicate definitions of SCG and SCH have been removed (lines 76, 80).

• **in figure 3, numbering of the subfigures are in capital letters instead of lowercase.**

The subfigures in Figure 4 (Figure 3 in the original manuscript) are labeled in lowercase letters which is consistent with the rest of the manuscript, however the subfigure labels in the figure caption have been changed from capital letters to lowercase (lines 289-292).

• **at line 290 and 291, the symbol epsilon is not consistent with that used in section 2 and Figures 2 and 5.**

The epsilon symbol present in original Lines 290 and 291 (now line 378) have been replaced with the epsilon symbol used in the rest of the manuscript to maintain consistency (ϵ).

Other minor edits have been made by the Authors to improve reading comprehension and/or grammar, and do not affect the results or conclusions of this study:

- Line 2: The title was changed to remove punctuation and replaced with “Spontaneous crack healing in calcite reveals the influence of dynamic strain evolution and surface chemistry”
- Line 33: a comma was removed
- Line 47: “Introduction” capitalized and heading number removed
- Line 48: “subcritical” was replaced with “spontaneous”
- Line 95: “offering” was changed to “offers”
- Line 96: “micron scale” was replaced with “micro-scale”
- Line 99-100: “uses in-situ Laue microdiffraction to investigate the time-dependent healing of cracks in calcite” was removed due to redundancy, and “shedding” was changed to “sheds”.
- Line 106: “Results” capitalized and heading number removed
- Line 107: Subheading number removed
- Line 108-109: “in a pre-notched piece of calcite” was removed as “A pre-notched, rectangular slice of calcite” was added to the prior sentence.
- Line 114: “The sample was then partially unloaded and” was added
- Line 116, 168, 170, 174, 245, 251, 275, 289, 314, 579, 601, 604: “pattern(s)” was replaced with “image(s)” for consistency
- Line 133: The color of the line representing the crack in Figure 2 was changed from purple to red for better visibility and the figure caption was updated accordingly
- Line 158: “cracktip_unloaded” was added
- Line 193: “y-direction” was replaced with “Y-direction” for consistency
- Line 195-196: “in the Y-direction” was added for clarity
- Line 196: “z-direction” was replaced with “Z-direction” for consistency
- Line 197: “parallel to the X-ray beam path and along the thickness of the sample” was added
- Line 198: “z-direction” was replaced with “Z-direction”, “dilatational” was replaced with “dilatational”
- Line 199: “y-direction” was replaced with “Y-direction”
- Line 200: “x-direction” was replaced with “X-direction”, and “parallel with the path of crack propagation” was added
- Line 204: “minimum” was added
- Line 206: “(ϵ_{yy})” was added
- Line 208-209: “This reflects the time-dependent accumulation of strain along the crack plane and simultaneous” was added
- Line 210 and 211: “values” was added
- Line 214: “the baseline-corrected” was added for clarity
- Line 217: “0.0279” was corrected to “0.2790”
- Line 223: “focusing” was replaced with “accumulation”

Lines 237-238: Words to clarify the (X,Y,Z) sample coordinate system and their orthogonality were added.

Line 239: “Horizontal and vertical scales are equivalent.” was added

Line 241: Subheading number removed

Line 245-246: “Trendlines correlated the measured average peak width value from each diffraction pattern to X-ray beam” was added

Line 247: “decreased” was changed to “decreasing”

Lines 248-249: “and the average peak width of each diffraction image was” was removed and replaced with “were used to calculate the contribution of peak width caused by the variable”

Line 255: “vertical” was removed

Line 261-262: “of the peak” and a reference to Kou et al. (2024) were added

Line 264: “ultimately” was added

Line 291: “Miller indices” was added

Line 294-295: “Intensity ranges from minimum (blue) to maximum (red).” was added

Line 300: “directions” was moved after “planes”

Line 301 (Table 2): “hkl” was added after “Peak” and “hkil” was added after “Slip system”

Line 307-308: “Select diffraction” and “(hkl)” were added

Line 309: Closing double apostrophes replaced the colon following “c”.

Line 310: “Intensity ranges from minimum (blue) to maximum (red).” was added

Line 313: Subheading number removed

Line 313: “mechanical testing” was removed and “X-ray diffraction data collection and removal of the calcite slide from the double-torsion device,” was added

Line 319-320: “or at a wavelength of ~6.2 um”, a reference to Seki et al. 2020, and “diffraction-limited” were added

Line 333: “the water” replaced “this”

Line 360” “persistent” was added

Line 368: References to Wan et al. 1990 and Wang et al. 2023 were added.

Line 373: “Discussion” capitalized and heading number removed

Line 376: “that became optically indistinguishable from the bulk following load removal” was added

Line 377: “In particular” was replaced with “Over time”

Line 378-379: “increased over time” and “became more pronounced” were removed and “increased in magnitude and localized closer to the location of the crack-air interface” was added

Lines 397-416: The paragraphs beginning with “Increases in diffraction peak width” and the following one starting with “Diffraction peak shape analysis identified” were moved up to improve flow

Lines 417-423: Due to formatting requirements, the Conclusions section was removed. Relevant and non-redundant information has been moved into the Discussion section, including paragraph starting with “Scanning infrared spectroscopy....”

Lines 497-507: Due to formatting requirements, the Conclusions section was removed. Relevant and non-redundant information has been moved into the Discussion section, including paragraph starting with “These findings indicate that the stress imparted”.

Line 551: Heading number and “Materials and” removed. “METHODS” capitalized to distinguish from subheadings.

Line 552: Subheading unitalicized, bolded, and colon removed

Line 554: The sample’s provenance and supplier was added: “of Brazilian provenance (sourced and”

Line 565: Subheading unitalicized, bolded, and colon removed

Line 573: “misplaced and” was added

Line 574-575: “due to inability to predict crack propagation path. This control scan was used” was added

Line 589: “and loaded” was removed

Line 590: “a” was added and “regions” was made singular

Line 593: “scans” was added

Line 594: “scan” was added and “unloaded” was removed

Line 598: Subheading unitalicized and bolded

Line 601: “at the ALS beamline 2.4” was added

Line 614: Subheading unitalicized and bolded

Line 626: “Data availability” capitalized and colon removed

Line 628: “Supplementary” was replaced with “Source”

Line 629: “from the authors” was added

Line 631: “References” capitalized

Line 898: “Acknowledgements” capitalized

Line 919: “Author Contributions” capitalized, colon removed

Line 928: “Competing interests” capitalized

Minor changes were made to the Source Data File that do not change the results or outcomes of this study:

- Tab names were updated to correspond to the revised figure numbers in the Manuscript and Supplementary Materials.
- “Table 1” was added to the name of tabs 1 and 2 to indicate the location of the data used to generate Table 1.
- A new tab (“Suppl. Table 1”) was added with the control scan source data used to generate Supplementary Table 1.